# HFIP-Promoted Bischler Indole Synthesis under Microwave Irradiation

**DOI:** 10.3390/molecules23123317

**Published:** 2018-12-14

**Authors:** Guangkai Yao, Zhi-Xiang Zhang, Cheng-Bei Zhang, Han-Hong Xu, Ri-Yuan Tang

**Affiliations:** 1Key Laboratory of Natural Pesticide and Chemical Biology, Ministry of Education, South China Agricultural University, Guangzhou 510642, China; yaoguangkai111@163.com (G.Y.); zdsys@scau.edu.cn (Z.-X.Z.); zhangcb0313@163.com (C.-B.Z.); 2State Key Laboratory for Conservation and Utilization of Subtropical Agro-Bioresources, South China Agricultural University, Guangzhou 510642, China; 3Department of Applied Chemistry, College of Materials and Energy, South China Agricultural University, Guangzhou 510642, China

**Keywords:** indole, cyclization, microwave synthesis

## Abstract

1,1,1,3,3,3-Hexafluoropropan-2-ol (HFIP) was found to be effective for the Bischler indole synthesis under microwave irradiation in the absence of a metal catalyst. Under the catalysis of HFIP, a wide range of α-amino arylacetones were successfully transformed into indole derivatives with moderate to good yields.

## 1. Introduction

Indole is an important structural unit that is present in many natural alkaloids [1,2,3]. Indole derivatives, such as indole-3-acetic acid (IAA) [4], physostigmine [5], amisulbrom [6], and fluvastatin [7], possess a large range of biological activities (Scheme 1A). The development of efficient and simple synthetic methods for indoles is highly desired [8,9,10]. The classical name reactions, such as Bischler indole synthesis [11,12], Fischer indole synthesis [13,14], Bartoli indole synthesis [15,16], have been widely used for indole synthesis. In recent years, efforts have been devoted to improve the Bischler indole synthesis [17,18,19,20,21,22,23]. Metal catalysts, including Rh [17], Ir [18], Ru [19], and Zn [20], have been applied to such transformations. However, most of these metal reagents are expensive, and the metal pollution is an inevitable problem in the preparation of indoles. α-Amino arylacetone, being used for indole synthesis, has multiple reactive sites [24,25,26]. In the presence of a metal catalyst, α-amino arylacetone was transformed into indolone [24] or *a*-ketoamide [25], and even underwent the cleavage of the C-N bond [26]. Thus, the development of metal-free and mild conditions for the selective transformation of α-amino arylacetone is highly desired. Chen and co-workers reported that NH_4_PF_6_ could promote the Bischler indole synthesis under metal-free conditions [27]. However, the preparation of NH_4_PF_6_ requires the use of corrosive NH_4_F and PCl_5_, which are harmful to the environment. We envision that a protic solvent would be able to promote the cyclodehydration of α-amino carbonyl compounds at the assistance of microwave irradiation. Because microwave synthetic technology often greatly improves the reaction efficiency and shortens the reaction time [28,29,30]. Herein we report a HFIP-promoted Bischler indole synthesis under microwave irradiation in the absence of any additive (Scheme 1B).

## 2. Results and Discussion

Our study began with the reaction of 2-(methylphenylamino)-1-phenyl-ethanone (**1a**) with different protic solvents under microwave irradiation (Table 1, entries 1–4). The reaction proceeded well in 1,1,1,3,3,3-hexafluoropropan-2-ol (HFIP) at 120 °C under microwave irradiation for 30 min, giving the desired product **2** in 76% yield (entry 1). CF_3_CH_2_OH was also effective for the cyclodehydration reaction, albeit with a 45% yield (entry 2). The reaction with *i*-PrOH or EtOH gave product **2** only 21% and 23% yield, respectively (entries 3 and 4). The reaction temperature significantly influenced the product yield. The product yield decreased when the reaction was conducted at a lower temperature (e.g., 51% yield at 80 °C and 72% yield at 100 °C (entries 5 and 6)). Next, the reaction time was examined. The product yield was increased to 88% by prolonging the reaction time to 40 min (entry 8). Whereas the reaction at 20 min reduced the product yield to 58% (entry 7). The product **2** was also obtained with 86% yield when the reaction was conducted under oil bath heating conditions for 16 h (entry 9). It is worth noting that the compound **2** is an excellent inhibitor of tubulin polymerization [31,32].

With the optimized reaction conditions in hand, the reaction scope was investigated (Table 2). Initially, substituents, including methyl, phenyl, methoxy, nitro, trifluoromethyl, cyano, bromo, chloro, and fluoro, on the benzene ring of arylethanone were investigated. All of these groups were well accepted to provide their corresponding products (**3**–**14**) in moderate to good yields. Most of these functional groups are readily modified to provide a wide range of indole derivatives. To our delight, substrates bearing a naphthalene, benzodioxepin, or benzofuran motif still performed well to provide their corresponding products (**15**–**17**) in moderate to good yields.

It is worth noting that products **18** and **19**, with an ester group, were also successfully prepared. The further modification of the ester group is able to provide diverse interesting indole derivatives. Subsequently, substituents on the benzene ring of aniline were evaluated. The results show that methoxy, bromo, and chloro groups were compatible with the reaction conditions, providing their corresponding products (**20**–**22**) in good yields. The reaction with 1-phenyl-2-(phenylamino)ethanone did not occur, suggesting that the *N*-methyl protecting group is essential for such a transformation (product **23**). To our delight, the benzyl protecting substrate also underwent the reaction well to give product **24** in 82% yield; whereas the reaction of *N*-Boc protecting substrate did not occur. Interestingly, the pharmaceutically important pyrrolo[3,2,1-*ij*]quinoline derivatives were also successfully synthesized by this simple protocol [33,34,35]. Numerous useful functional groups, such as methoxy, trifluoromethyl, bromo, chloro, and fluoro groups, were well tolerated (products **25**–**29**). For example, halo-substituted compounds **27** and **28** were synthetically useful for further modifications by cross-coupling reactions. Both naphthalene and benzofuran moieties were also accepted to afford products **31** and **32** in good yields. The ester group of product **33** is attractive to the medicinal community. 1-[Methyl(phenyl)amino]butan-2-one still underwent the reaction smoothly to provide the desired product **34** in 90% yield.

According to the Bischler reaction mechanism [11,12] and the present results, a possible reaction pathway was illustrated, as shown in Scheme 2. HFIP may play the role of an acid that activates the carbonyl for the Friedel-Crafts cyclization to produce an intermediate **A**. The HFIP anion facilitates the elimination of the hydrogen atom to produce an intermediate **B**. The hydroxyl of the intermediate **B** accepts a proton from HFIP to form an intermediate **C**, followed by a dehydration process to provide an intermediate **D**. Finally, the intermediate **D** undergoes a deprotonation process to afford the desired product. In the reaction, microwave irradiation may promote the electron flow of α-amino arylacetone, and greatly improve the reaction efficiency.

## 3. Materials and Methods

### 3.1. General Information

NMR spectroscopy were performed on a Bruker Avance 500 instrument (Bruker, Billerica, MA, USA) (500 MHz for ^1^H, 126 MHz for ^13^C-NMR spectroscopy), using CDCl_3_ and DMSO-*d*_6_ as the solvent, and calibrated using residual deuterated solvents as an internal reference (CDCl_3_: δ 7.26 ppm for ^1^H-NMR, δ 77.16 ppm for ^13^C-NMR; DMSO-*d*_6_: δ 2.50 ppm for ^1^H-NMR, δ = 39.52 ppm for ^13^C-NMR). The following abbreviations (or combinations thereof) were used to explain multiplicities: s = singlet, d = doublet, t = triplet, q = quartet, m = multiplet. Mass spectra were measured on an Agilent GC-MS-5975C Plus spectrometer (EI) (Agilent, Santa Clara, CA, USA). LC-MS (ESI) analysis was measured on an AB Sciex API3200 (AB SCIEX, Framingham, MA, USA). HRMS (ESI) analyses was measured on a Thermo Scientific LTQ Orbitrap XL instrument (Thermo Scientific, Waltham, MA, USA). Microwave experiments were conducted in a CEM Discover single-mode instrument (CEM Corporation, Matthews, NC, USA) using the internal probe.

### 3.2. General Procedure for the Synthesis of Indoles and Pyrrolo[3,2,1-ij]quinolones

A mixture of α-amino arylethanone (**1a**) (0.6 mmol) in HFIP (3 mL) was sealed in a pressure vessel tube, and was stirred at 120 °C under microwave irradiation for 40 min. After the reaction finished, the crude reaction mixture was diluted with EtOAc (5 mL), and filtered through a short pad of celite. The sealed tube and celite pad were washed with an additional 20 mL of EtOAc. The filtrate was concentrated in vacuo, and the resulting residue was purified by flash column chromatography using hexanes and EtOAc as the eluent (NMR spectra for all compounds shown in Appendix A).

*1-Methyl-3-phenyl-1H-indole* (**2**). Yellow solid. 87% yield (108 mg). M.P.: 65–67 °C. ^1^H-NMR (500 MHz, CDCl_3_) δ 7.94 (d, *J* = 8.0 Hz, 1H), 7.64 (dd, *J* = 8.3, 1.3 Hz, 2H), 7.42 (t, *J* = 7.8 Hz, 2H), 7.34 (d, *J* = 8.2 Hz, 1H), 7.28–7.22 (m, 2H), 7.22–7.13 (m, 2H), 3.79 (s, 3H). ^13^C-NMR (126 MHz, CDCl_3_) δ 137.5, 135.6, 128.7, 127.3, 126.5, 126.1, 125.7, 121.9, 119.9, 119.8, 116.7, 109.5, 32.8; LRMS (EI, 70 Ev) *m*/*z* (%): 207 (M^+^, 100).

*1-Methyl-3-(p-tolyl)-1H-indole* (**3**). Yellow solid. 83% yield (110 mg). M.P.: 66–68 °C. ^1^H-NMR (500 MHz, CDCl_3_) δ 7.98 (d, *J* = 8.0 Hz, 1H), 7.60 (d, *J* = 8.0 Hz, 2H), 7.39 (d, *J* = 8.2 Hz, 1H), 7.35–7.27 (m, 4H), 7.25–7.18 (m, 3H), 3.84 (s, 3H), 2.44 (s, 3H). ^13^C-NMR (126 MHz, CDCl_3_) δ 137.4, 135.2, 132.7, 129.4, 127.2, 126.2, 121.9, 119.9, 119.7, 116.6, 109.4, 32.8, 21.1; LRMS (EI, 70 Ev) *m*/*z* (%): 221 (M^+^, 100).

*3-([1,1′-Biphenyl]-4-yl)-1-methyl-1H-indole* (**4**). Pale yellow soild. 66% yield (112 mg). M.P.: 131–132 °C. ^1^H-NMR (500 MHz, CDCl_3_) δ 8.09 (d, *J* = 7.9 Hz, 1H), 7.82 (d, *J* = 8.1 Hz, 2H), 7.79–7.72 (m, 4H), 7.54 (t, *J* = 7.7 Hz, 2H), 7.47–7.40 (m, 2H), 7.38 (t, *J* = 7.6 Hz, 1H), 7.34–7.27 (m, 2H), 3.88 (s, 3H). ^13^C-NMR (126 MHz, CDCl_3_) δ 141.0, 138.3, 137.5, 134.7, 128.7, 127.5, 127.4, 127.0, 126.9, 126.6, 126.1, 122.0, 119.9, 116.2, 109.5, 32.8; HRMS (ESI) for C_21_H_18_N (M + H^+^): calcd. 284.1434, found 284.1436.

*3-(4-Methoxyphenyl)-1-methyl-1H-indole* (**5**). Yellow solid. 77% yield (109 mg). M.P.: 97–99 °C. ^1^H-NMR (500 MHz, CDCl_3_) δ 7.93 (d, *J* = 8.0 Hz, 1H), 7.63–7.56 (m, 2H), 7.38 (d, *J* = 8.2 Hz, 1H), 7.33–7.27 (m, 1H), 7.21 (td, *J* = 7.5, 7.1, 1.0 Hz, 1H), 7.17 (s, 1H), 7.07–6.98 (m, 2H), 3.88 (s, 3H), 3.84 (s, 3H).^13^C-NMR (126 MHz, CDCl_3_) δ 157.9, 137.3, 128.4, 128.2, 126.2, 125.9, 121.8, 119.8, 119.6, 116.4, 114.2, 109.4, 55.3, 32.7; LRMS (EI, 70 Ev) *m*/*z* (%): 237 (M^+^, 100).

*1-Methyl-3-(4-nitrophenyl)-1H-indole* (**6**) [36]. Light yellow solid. 72% yield (109 mg). M.P.: 139–140 °C. ^1^H-NMR (500 MHz, CDCl_3_) δ 8.38–8.16 (m, 2H), 7.96 (d, *J* = 8.0 Hz, 1H), 7.84–7.71 (m, 2H), 7.42 (d, *J* = 8.1 Hz, 2H), 7.35 (t, *J* = 7.6 Hz, 1H), 7.31–7.24 (m, 1H), 3.88 (s, 3H). ^13^C-NMR (126 MHz, CDCl_3_) δ 145.2, 142.8, 137.8, 128.3, 126.6, 125.6, 124.3, 122.7, 121.0, 119.6, 114.6, 110.0, 33.1; LRMS (EI, 70 Ev) *m*/*z* (%): 252 (M^+^, 100).

*1-Methyl-3-(4-(trifluoromethyl)phenyl)-1H-indole* (**7**) [27]. White solid. 65% yield (107 mg). M.P.: 81–82 °C. ^1^H-NMR (500 MHz, CDCl_3_) δ 7.95 (d, *J* = 8.0 Hz, 1H), 7.76 (d, *J* = 8.1 Hz, 2H), 7.68 (d, *J* = 8.1 Hz, 2H), 7.40 (d, *J* = 8.2 Hz, 1H), 7.35–7.28 (m, 2H), 7.27–7.17 (m, 1H), 3.86 (s, 3H). ^13^C-NMR (126 MHz, CDCl_3_) δ 139.4, 137.6, 127.6, 127.3, 127.0, 125.9, 125.7 (q, *J* = 3.8 Hz), 124.6 (q, *J* = 270.1 Hz), 122.4, 120.4, 119.7, 115.4, 109.8, 33.0; LRMS (EI, 70 Ev) *m*/*z* (%): 275 (M^+^, 100).

*4-(1-Methyl-1H-indol-3-yl)benzonitrile* (**8**). Light yellow solid. 73% yield (102 mg). M.P.: 146–148 °C. ^1^H-NMR (500 MHz, CDCl_3_) δ 7.96 (d, *J* = 8.0 Hz, 1H), 7.76 (d, *J* = 8.1 Hz, 2H), 7.70 (d, *J* = 8.2 Hz, 2H), 7.43 (d, *J* = 8.2 Hz, 1H), 7.40–7.32 (m, 2H), 7.29 (t, *J* = 7.5 Hz, 1H), 3.88 (s, 3H). ^13^C-NMR (126 MHz, CDCl_3_) δ 140.6, 137.6, 132.5, 127.8, 126.9, 125.5, 122.5, 120.7, 119.5, 119.4, 114.8, 109.9, 108.3, 33.0; LRMS (EI, 70 Ev) *m*/*z* (%): 232 (M^+^, 100).

*3-(4-Bromophenyl)-1-methyl-1H-indole* (**9**) [36]. Yellow solid. 82% yield (140 mg). M.P.: 108–109 °C. ^1^H-NMR (500 MHz, CDCl_3_) δ 7.90 (d, *J* = 8.0 Hz, 1H), 7.62–7.49 (m, 4H), 7.38 (d, *J* = 8.2 Hz, 1H), 7.35–7.28 (m, 1H), 7.25–7.17 (m, 2H), 3.84 (s, 3H). ^13^C-NMR (126 MHz, CDCl_3_) δ 137.5, 134.6, 131.8, 128.7, 126.6, 125.9, 122.1, 120.1, 119.6, 119.2, 115.5, 109.6, 32.8; LRMS (EI, 70 Ev) *m*/*z* (%): 287 (M^+^, 100), 285 (M^+^, 100).

*3-(4-Chlorophenyl)-1-methyl-1H-indole* (**10**). Light brown solid. 76% yield (110 mg). M.P.: 97–99 °C. ^1^H-NMR (500 MHz, CDCl_3_) δ 7.90 (d, *J* = 8.0 Hz, 1H), 7.67–7.53 (m, 2H), 7.45–7.35 (m, 3H), 7.33–7.29 (m, 1H), 7.25–7.19 (m, 2H), 3.84 (s, 3H). ^13^C-NMR (126 MHz, CDCl_3_) δ 137.5, 134.1, 131.2, 128.8, 128.4, 126.6, 125.9, 122.1, 120.1, 119.6, 115.5, 109.6, 32.9; LRMS (EI, 70 Ev) *m*/*z* (%): 241 (M^+^, 100).

*1-Methyl-3-(3-nitrophenyl)-1H-indole* (**11**). Orange solid. 56% yield (85 mg). M.P.: 108–109 °C. ^1^H-NMR (500 MHz, CDCl_3_) δ 8.49 (t, *J* = 2.0 Hz, 1H), 8.08 (ddd, *J* = 8.2, 2.3, 1.1 Hz, 1H), 7.98 (dd, *J* = 7.8, 1.4 Hz, 1H), 7.94 (d, *J* = 7.9 Hz, 1H), 7.57 (t, *J* = 7.9 Hz, 1H), 7.41 (d, *J* = 8.2 Hz, 1H), 7.38–7.30 (m, 2H), 7.26 (t, *J* = 7.5 Hz, 2H), 3.87 (s, 2H). ^13^C-NMR (126 MHz, CDCl_3_) δ 148.7, 137.5, 137.5, 132.6, 129.5, 127.4, 125.6, 122.5, 121.4, 120.6, 120.1, 119.3, 114.3, 109.8, 33.0; LRMS (EI, 70 Ev) *m*/*z* (%): 252 (M^+^, 100); HRMS (ESI) for C_15_H_13_N_2_O_2_ (M + H^+^): calcd. 253.0899, found 253.0894.

*3-(1-Methyl-1H-indol-3-yl)benzonitrile* (**12**). Yellow solid. 61% yield (85 mg). M.P.: 121–124 °C. ^1^H-NMR (500 MHz, CDCl_3_) δ 7.98–7.80 (m, 3H), 7.58–7.46 (m, 2H), 7.40 (d, *J* = 8.2 Hz, 1H), 7.36–7.30 (m, 1H), 7.29–7.18 (m, 2H), 3.86 (s, 3H). ^13^C-NMR (126 MHz, CDCl_3_) δ 137.5, 137.0, 131.2, 130.3, 129.5, 128.8, 127.1, 125.6, 122.4, 120.5, 119.3, 119.1, 114.4, 112.8, 109.8, 33.0; LRMS (EI, 70 Ev) *m*/*z* (%): 232 (M^+^, 100); HRMS (ESI) for C_16_H_13_N_2_ (M + H^+^): calcd. 232.2860, found 232.2868.

*3-(3,4-Dimethoxyphenyl)-1-methyl-1H-indole* (**13**). Yellow oil. 76% yield (122 mg). ^1^H-NMR (500 MHz, DMSO-*d*_6_) δ 7.86 (d, *J* = 8.0 Hz, 1H), 7.60 (s, 1H), 7.47 (d, *J* = 8.3 Hz, 1H), 7.26–7.15 (m, 3H), 7.15–7.10 (m, 1H), 7.01 (d, *J* = 8.2 Hz, 1H), 3.84 (s, 3H), 3.82 (s, 3H), 3.78 (s, 3H). ^13^C-NMR (126 MHz, DMSO-*d*_6_) δ 149.5, 147.4, 137.6, 128.8, 127.5, 125.9, 121.8, 120.0, 119.7, 119.0, 115.4, 113.0, 111.1, 110.5, 56.1, 55.9, 32.9; LRMS (EI, 70 Ev) *m*/*z* (%): 267 (M^+^, 100); HRMS (ESI) for C_18_H_18_NO_2_ (M + H^+^): calcd. 268.1332, found 268.1335.

*3-(3,4-Difluorophenyl)-1-methyl-1H-indole* (**14**) [27]. Yellow oil. 73% yield (106 mg).^1^H-NMR (500 MHz, CDCl_3_) δ 7.87 (d, *J* = 8.0 Hz, 1H), 7.48–7.40 (m, 1H), 7.38 (d, *J* = 8.2 Hz, 1H), 7.36–7.28 (m, 2H), 7.25–7.17 (m, 3H), 3.85 (s, 3H). ^13^C-NMR (126 MHz, CDCl_3_) δ151.5 (d, *J* = 12.8 Hz), 149.6 (t, *J* = 13.0 Hz), 147.7 (d, *J* = 12.8 Hz), 137.4, 132.8 (dd, *J* = 6.6, 3.8 Hz), 126.7, 125.8, 122.9 (dd, *J* = 5.8, 3.3 Hz), 122.3, 120.2, 119.4, 117.4 (d, *J* = 17.4 Hz), 115.8 (d, *J* = 17.4 Hz), 114.9, 109.7, 32.9; LRMS (EI, 70 Ev) *m*/*z* (%): 243 (M^+^, 100).

*1-Methyl-3-(naphthalen-2-yl)-1H-indole* (**15**). Colorless oil. 82% yield (126 mg). ^1^H-NMR (500 MHz, CDCl_3_) δ 8.20 (s, 1H), 8.17 (d, *J* = 7.9 Hz, 1H), 8.00–7.90 (m, 3H), 7.87 (dd, *J* = 8.5, 1.7 Hz, 1H), 7.60–7.49 (m, 2H), 7.44 (d, *J* = 8.1 Hz, 1H), 7.42–7.37 (m, 1H), 7.36 (s, 1H), 7.35–7.31 (m, 1H), 3.85 (s, 3H). ^13^C-NMR (126 MHz, CDCl_3_) δ 137.5, 134.0, 133.1, 131.9, 128.2, 127.7, 127.6, 126.9, 126.4, 126.2, 126.0, 125.0, 124.8, 122.0, 120.0, 120.0, 116.5, 109.6, 32.8; LRMS (EI, 70 Ev) *m*/*z* (%): 257 (M^+^, 100).

*3-(3,4-Dihydro-2H-benzo[b][1,4]dioxepin-7-yl)-1-methyl-1H-indole* (**16**). Yellow oil. 55% yield (92 mg). ^1^H-NMR (500 MHz, CDCl_3_) δ 7.98 (dd, *J* = 8.0, 1.0 Hz, 1H), 7.39 (d, *J* = 8.3 Hz, 1H), 7.35 (d, *J* = 2.2 Hz, 1H), 7.34–7.30 (m, 1H), 7.28 (dd, *J* = 8.5, 1.8 Hz, 1H), 7.27–7.20 (m, 1H), 7.21 (s, 1H), 7.11 (d, *J* = 8.1 Hz, 1H), 4.31 (dt, *J* = 10.9, 5.6 Hz, 4H), 3.84 (s, 3H), 2.34–2.17 (m, 2H). ^13^C-NMR (126 MHz, CDCl_3_) δ 151.3, 149.3, 137.3, 131.2, 126.2, 126.0, 122.1, 121.8, 121.8, 120.1, 119.8, 119.7, 115.8, 109.4, 70.6, 70.5, 32.7, 32.0; LRMS (EI, 70 Ev) *m*/*z* (%): 279 (M^+^, 100); HRMS (ESI) for C_18_H_18_NO_2_ (M + H^+^): calcd. 280.1259, found 280.1263.

*3-(Benzofuran-2-yl)-1-methyl-1H-indole* (**17**) [27]. Yellow oil. 47% yield (70 mg). ^1^H-NMR (500 MHz, CDCl_3_) δ 8.06 (d, *J* = 7.8 Hz, 1H), 7.61 (s, 1H), 7.60–7.55 (m, 1H), 7.54–7.49 (m, 1H), 7.39 (d, *J* = 7.9 Hz, 1H), 7.37–7.27 (m, 2H), 7.27–7.20 (m, 2H), 6.91 (s, 1H), 3.85 (s, 3H). ^13^C-NMR (126 MHz, CDCl_3_) δ 153.8, 152.9, 137.4, 129.9, 127.5, 125.0, 123.0, 122.6, 122.5, 120.6, 120.3, 120.0, 110.5, 109.7, 106.9, 99.0, 33.0; LRMS (EI, 70 Ev) *m*/*z* (%): 247 (M^+^, 100).

*Ethyl 1-methyl-1H-indole-3-carboxylate* (**18**) [37]. Yellow oil. 45% yield (55 mg). ^1^H-NMR (500 MHz, DMSO-*d*_6_) δ 8.10 (s, 1H), 8.02 (dd, *J* = 7.1, 1.3 Hz, 1H), 7.53 (d, *J* = 8.1 Hz, 1H), 7.30–7.20 (m, 2H), 4.28 (q, *J* = 7.2 Hz, 2H), 3.85 (s, 3H), 1.33 (t, *J* = 7.1 Hz, 3H). ^13^C-NMR (126 MHz, DMSO-*d*_6_) δ 164.5, 137.4, 136.5, 126.5, 122.8, 121.9, 121.0, 111.2, 105.9, 59.4, 33.4, 14.9; LRMS (EI, 70 Ev) *m*/*z* (%): 203 (M^+^, 100).

*Ethyl 2-(1-methyl-1H-indol-3-yl)acetate* (**19**) [27]. Yellow oil. 72% yield (94 mg). ^1^H-NMR (500 MHz, CDCl_3_) δ 7.64 (d, *J* = 7.9 Hz, 1H), 7.32 (d, *J* = 8.2 Hz, 1H), 7.28–7.22 (m, 1H), 7.18–7.12 (m, 1H), 7.06 (s, 1H), 4.19 (q, *J* = 7.1 Hz, 2H), 3.78 (s, 2H), 3.77 (s, 3H), 1.29 (t, *J* = 7.1 Hz, 3H). ^13^C-NMR (126 MHz, CDCl_3_) δ 172.1, 136.8, 127.7, 127.6, 121.7, 119.0, 118.9, 109.2, 106.9, 60.7, 32.6, 31.3, 14.2; LRMS (EI, 70 Ev) *m*/*z* (%): 217 (M^+^, 100).

*5-Methoxy-1-methyl-3-phenyl-1H-indole* (**20**). Light yellow solid. 78% yield (111 mg). M.P.: 80–81 °C. ^1^H-NMR (500 MHz, CDCl_3_) δ 7.65 (d, *J* = 7.1 Hz, 2H), 7.46 (t, *J* = 7.7 Hz, 2H), 7.41 (d, *J* = 2.4 Hz, 1H), 7.32–7.24 (m, 1H), 7.21 (s, 1H), 6.96 (dd, *J* = 8.8, 2.4 Hz, 1H), 3.89 (s, 3H), 3.82 (s, 3H).^13^C-NMR (126 MHz, CDCl_3_) δ 154.6, 135.8, 132.9, 128.7, 127.2, 127.1, 126.4, 125.6, 116.3, 112.2, 110.2, 101.8, 56.0, 33.0; LRMS (EI, 70 Ev) *m*/*z* (%): 237 (M^+^, 100).

*5-Bromo-1-methyl-3-phenyl-1H-indole* (**21**) [36]. Yellow oil. 65% yield (111 mg). ^1^H-NMR (500 MHz, CDCl_3_) δ 8.06 (d, *J* = 1.8 Hz, 1H), 7.61 (d, *J* = 7.2 Hz, 2H), 7.45 (t, *J* = 7.7 Hz, 2H), 7.36 (dd, *J* = 8.7, 1.7 Hz, 1H), 7.30 (t, *J* = 7.4 Hz, 1H), 7.22 (d, *J* = 8.6 Hz, 2H), 3.81 (s, 3H). ^13^C-NMR (126 MHz, CDCl_3_) δ 136.1, 134.8, 128.8, 127.8, 127.5, 127.3, 126.0, 124.8, 122.4, 116.4, 113.4, 111.0, 33.0; LRMS (EI, 70 Ev) *m*/*z* (%): 287 (M^+^, 100), 285 (M^+^, 100).

*5-Chloro-1-methyl-3-phenyl-1H-indole* (**22**) [27]. Yellow soild. 77% yield (111 mg). M.P.: 87–89 °C. ^1^H-NMR (500 MHz, CDCl_3_) δ 7.90 (d, *J* = 1.8 Hz, 1H), 7.61 (dd, *J* = 8.2, 1.2 Hz, 2H), 7.45 (t, *J* = 7.8 Hz, 2H), 7.33–7.27 (m, 2H), 7.25–7.20 (m, 2H), 3.82 (s, 3H). ^13^C-NMR (126 MHz, CDCl_3_) δ 135.8, 134.9, 128.8, 127.7, 127.2, 127.1, 126.0, 125.8, 122.2, 119.3, 116.5, 110.5, 33.0; LRMS (EI, 70 Ev) *m*/*z* (%): 241 (M^+^, 100).

*1-benzyl-3-phenyl-1H-indole* (**24**). Yellow oil. 82% yield (146 mg). ^1^H-NMR (500 MHz, CDCl_3_) δ 8.02 (d, *J* = 7.3 Hz, 1H), 7.71 (dd, *J* = 8.2, 1.1 Hz, 2H), 7.47 (t, *J* = 7.7 Hz, 2H), 7.39–7.28 (m, 6H), 7.28–7.23 (m, 2H), 7.21 (d, *J* = 6.8 Hz, 2H), 5.39 (s, 2H). ^13^C-NMR (126 MHz, CDCl_3_) δ 137.3, 137.2, 135.6, 128.9, 128.8, 127.8, 127.4, 127.0, 126.5, 126.0, 125.9, 122.2, 120.2, 120.1, 117.4, 110.1, 50.2; LRMS (EI, 70 Ev) *m*/*z* (%): 283 (M^+^, 100).

*1-(4-Methoxyphenyl)-5,6-dihydro-4H-pyrrolo[3,2,1-ij]quinoline* (**25**) [27]. Yellow solid. 82% yield (129 mg). M.P.: 127–129 °C. ^1^H-NMR (500 MHz, CDCl_3_) δ 7.74 (d, *J* = 8.0 Hz, 1H), 7.68–7.52 (m, 2H), 7.22 (s, 1H), 7.11 (dd, *J* = 8.0, 7.1 Hz, 1H), 7.04–6.87 (m, 3H), 4.99–3.96 (m, 2H), 3.87 (s, 3H), 3.04 (t, *J* = 6.1 Hz, 2H), 2.46–2.03 (m, 2H). ^13^C-NMR (126 MHz, CDCl_3_) δ 157.7, 134.8, 128.8, 127.9, 123.7, 123.0, 121.9, 120.1, 118.8, 117.4, 116.2, 114.2, 55.3, 44.1, 24.7, 22.8; LRMS (EI, 70 Ev) *m*/*z* (%): 263 (M^+^, 100).

*1-[4-(Trifluoromethyl)phenyl]-5,6-dihydro-4H-pyrrolo[3,2,1-ij]quinoline* (**26**) [27]. Yellow oil. 81% yield (146 mg). ^1^H-NMR (500 MHz, CDCl_3_) δ 7.84–7.74 (m, 3H), 7.68 (d, *J* = 8.1 Hz, 2H), 7.37 (s, 1H), 7.16 (dd, *J* = 8.0, 7.1 Hz, 1H), 7.02 (dd, *J* = 7.1, 1.1 Hz, 1H), 4.50–3.87 (m, 2H), 3.05 (t, *J* = 6.1 Hz, 2H), 2.58–2.10 (m, 2H). ^13^C-NMR (126 MHz, CDCl_3_) δ 139.9, 135.0, 127.8, 127.1 (q, *J* = 32.4 Hz), 126.5, 125.6 (q, *J* = 3.8 Hz), 124.5, 124.4 (q, *J* = 269.9 Hz), 123.5, 122.3, 120.9, 119.4, 117.3, 115.1, 44.3, 24.6, 22.7; LRMS (EI, 70 Ev) *m*/*z* (%): 301 (M^+^, 100).

*1-(4-Bromophenyl)-5,6-dihydro-4H-pyrrolo[3,2,1-ij]quinoline* (**27**) [27]. Yellow solid. 83% yield (155 mg). M.P.: 156–158 °C.^1^H-NMR (500 MHz, CDCl_3_) δ 7.71 (d, *J* = 7.6 Hz, 1H), 7.60–7.44 (m, 4H), 7.29 (s, 1H), 7.12 (dd, *J* = 8.0, 7.1 Hz, 1H), 7.04–6.92 (m, 1H), 4.29–4.13 (m, 2H), 3.03 (t, *J* = 6.1 Hz, 2H), 2.42–2.12 (m, 2H). ^13^C-NMR (126 MHz, CDCl_3_) δ 135.1, 134.9, 131.7, 128.2, 123.8, 123.4, 122.2, 120.6, 119.1, 118.8, 117.2, 115.3, 44.2, 24.6, 22.7; LRMS (EI, 70 Ev) *m*/*z* (%): 313 (M^+^, 100), 311 (M^+^, 100).

*1-(4-Chlorophenyl)-5,6-dihydro-4H-pyrrolo[3,2,1-ij]quinoline* (**28**) [27]. Yellow solid. 85% yield (136 mg). M.P.: 126–128 °C. ^1^H-NMR (500 MHz, CDCl_3_) δ 7.73 (d, *J* = 8.0 Hz, 1H), 7.65–7.57 (m, 2H), 7.45–7.35 (m, 2H), 7.28 (s, 1H), 7.17–7.10 (m, 1H), 7.00 (dd, *J* = 7.0, 1.0 Hz, 1H), 4.33–4.04 (m, 2H), 3.04 (t, *J* = 6.1 Hz, 2H), 2.55–1.99 (m, 2H). ^13^C-NMR (126 MHz, CDCl_3_) δ 134.9, 134.7, 130.9, 128.8, 127.9, 123.8, 123.5, 122.1, 120.6, 119.1, 117.2, 115.3, 44.2, 24.7, 22.7; LRMS (EI, 70 Ev) *m*/*z* (%): 267 (M^+^, 100).

*1-(4-Fluorophenyl)-5,6-dihydro-4H-pyrrolo[3,2,1-ij]quinoline* (**29**) [27]. Yellow oil. 87% yield (131 mg). ^1^H-NMR (500 MHz, CDCl_3_) δ 7.71 (d, *J* = 8.1 Hz, 1H), 7.67–7.60 (m, 2H), 7.24 (s, 1H), 7.17–7.08 (m, 3H), 6.99 (d, *J* = 7.0 Hz, 1H), 4.33–3.95 (m, 2H), 3.04 (t, *J* = 6.1 Hz, 2H), 2.51–2.08 (m, 2H). ^13^C-NMR (126 MHz, CDCl_3_) δ161.14 (d, *J* = 243.9 Hz), 134.87, 132.19 (d, *J* = 3.2 Hz), 128.19 (d, *J* = 7.7 Hz), 123.60, 123.49, 122.07, 120.42, 119.03, 117.19, 115.60 (d, *J* = 3.8 Hz), 115.42, 44.17, 24.68, 22.77; LRMS (EI, 70 Ev) *m*/*z* (%): 251 (M^+^, 100).

*3-(5,6-Dihydro-4H-pyrrolo[3,2,1-ij]quinolin-1-yl)benzonitrile* (**30**). Yellow oil. 71% yield (110 mg). ^1^H-NMR (500 MHz, CDCl_3_) δ 7.95 (d, *J* = 1.2 Hz, 1H), 7.93–7.88 (m, 1H), 7.73 (d, *J* = 8.0 Hz, 1H), 7.54–7.43 (m, 2H), 7.33 (s, 1H), 7.16 (dd, *J* = 8.1, 7.0 Hz, 1H), 7.02 (d, *J* = 6.9 Hz, 1H), 4.59–3.87 (m, 2H), 3.04 (t, *J* = 6.1 Hz, 2H), 2.42–2.13 (m, 2H).^13^C-NMR (126 MHz, CDCl_3_) δ 137.5, 134.9, 130.6, 129.7, 129.4, 128.5, 124.3, 123.2, 122.3, 121.0, 119.4, 119.2, 116.9, 114.2, 112.7, 44.3, 24.6, 22.6; HRMS (ESI) for C_18_H_16_N_2_ (M + H^+^): calcd. 259.1230, found 259.1231.

*1-(Naphthalen-2-yl)-5,6-dihydro-4H-pyrrolo[3,2,1-ij]quinoline* (**31**) [27]. White solid. 78% yield (132 mg). M.P.: 126–128 °C. ^1^H-NMR (500 MHz, CDCl_3_) δ 8.14 (s, 1H), 7.92–7.84 (m, 3H), 7.86–7.78 (m, 2H), 7.47 (ddd, *J* = 8.1, 6.7, 1.2 Hz, 1H), 7.45–7.38 (m, 2H), 7.15 (dd, *J* = 7.9, 7.1 Hz, 1H), 7.00 (dd, *J* = 7.0, 1.0 Hz, 1H), 4.35–4.05 (m, 2H), 3.04 (t, *J* = 6.1 Hz, 2H), 2.39–2.14 (m, 2H). ^13^C-NMR (126 MHz, CDCl_3_) δ 135.1, 134.1, 133.7, 131.8, 128.1, 127.7, 127.6, 126.0, 126.0, 124.9, 124.3, 124.2, 123.8, 122.1, 120.5, 119.1, 117.6, 116.4, 44.2, 24.7, 22.8; LRMS (EI, 70 Ev) *m*/*z* (%): 283 (M^+^, 100).

*1-(Benzofuran-2-yl)-5,6-dihydro-4H-pyrrolo[3,2,1-ij]quinoline* (**32**) [27]. Yellow solid. 71% yield (116 mg). M.P.: 141–143 °C.^1^H-NMR (500 MHz, CDCl_3_) δ 7.82 (d, *J* = 8.0 Hz, 1H), 7.63 (s, 1H), 7.60–7.54 (m, 1H), 7.53–7.46 (m, 1H), 7.27–7.16 (m, 3H), 7.03 (dd, *J* = 7.1, 1.1 Hz, 1H), 6.89 (d, *J* = 0.9 Hz, 1H), 4.45–3.88 (m, 2H), 3.04 (t, *J* = 6.1 Hz, 2H), 2.50–2.14 (m, 2H). ^13^C-NMR (126 MHz, CDCl_3_) δ 153.8, 153.5, 134.8, 130.0, 124.6, 122.8, 122.8, 122.6, 122.3, 121.0, 119.9, 119.5, 117.8, 110.5, 106.9, 98.7, 44.4, 24.6, 22.8; LRMS (EI, 70 Ev) *m*/*z* (%): 273 (M^+^, 100).

*Ethyl 5,6-dihydro-4H-pyrrolo[3,2,1-ij]quinoline-1-carboxylate* (**33**) [27]. Yellow oil. 66% yield (96 mg). ^1^H-NMR (500 MHz, CDCl_3_) δ 7.49–7.36 (m, 1H), 7.13–7.00 (m, 2H), 6.93 (dd, *J* = 7.1, 1.1 Hz, 1H), 4.19 (q, *J* = 7.1 Hz, 2H), 4.17–4.08 (m, 2H), 3.78 (d, *J* = 0.8 Hz, 2H), 3.00 (t, *J* = 6.1 Hz, 2H), 2.33–2.05 (m, 2H), 1.29 (t, *J* = 7.1 Hz, 3H). ^13^C-NMR (126 MHz, CDCl_3_) δ 172.2, 134.3, 125.2, 124.9, 121.6, 119.5, 118.5, 116.5, 106.8, 60.6, 43.9, 31.7, 24.6, 22.8, 14.2; LRMS (EI, 70 Ev) *m*/*z* (%): 243 (M^+^, 100).

*1-Ethyl-5,6-dihydro-4H-pyrrolo[3,2,1-ij]quinoline* (**34**) [27]. Yellow oil. 90% yield (100 mg). ^1^H-NMR (500 MHz, CDCl_3_) δ 7.43 (d, *J* = 7.9 Hz, 1H), 7.11–6.95 (m, 1H), 6.91 (dd, *J* = 7.0, 1.2 Hz, 1H), 6.87 (s, 1H), 4.28–3.89 (m, 2H), 2.99 (t, *J* = 6.1 Hz, 2H), 2.87–2.65 (m, 2H), 2.45–2.05 (m, 2H), 1.37–1.31 (m, 3H). ^13^C-NMR (126 MHz, CDCl_3_) δ 134.6, 125.1, 122.6, 121.5, 118.9, 118.2, 117.4, 116.6, 43.8, 24.7, 23.0, 18.7, 14.9; LRMS (EI, 70 Ev) *m*/*z* (%): 185 (M^+^, 100).

## 4. Conclusions

In summary, HFIP was found to be effective for Bischler indole synthesis under microwave irradiation in the absence of metal catalyst and additive. A variety of α-amino arylacetones were transformed into indole derivatives under the catalysis of HFIP. Interestingly, the pharmaceutically important pyrrolo[3,2,1-*ij*]quinoline derivatives were also successfully prepared by this simple protocol. This practical synthetic method has several advantages: Metal-free and additive-free conditions, H_2_O is the only by-product, and HFIP is readily recovered by rotary distillation.

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
