# Peer review of "HFIP-Promoted Bischler Indole Synthesis under Microwave Irradiation"

_molecules, 2018, doi:10.3390/molecules23123317_

Round 1
Reviewer 1 Report
The authors describe the synthesis of indole derivatives without metal catalyst under microwave irradiation. The work is well planned, the results are supported with the adequate experimental data and the proposed mechanism is reasonable.
The only restriction of the scope of the reaction is the necessity to have an alkyl group on the nitrogen atom (typically a methyl group). The N-methyl deprotection seems to me not so easy especially with a metal free approach. As a minimum, some methods must be added as example from the literature. More interesting should be to synthesize a compound bearing a protecting group easier to remove such as a benzyl moiety. That will be a huge improvement in the quality of the paper.
Some minor Errors and corrections must be addressed:
Line 31: one space is missing between “of” and “α”
Scheme 1. A. durgs => drugs
One carbon atoms is missing in the NMR description of the compounds 2 and 3.
There is a mistake in the 1H NMR integration value of two signals of the compound 2 at 3.84 and 2.44 ppm. That should be 3 instead of 4.
To conclude, if the authors can improve the “N-methyl deprotection part”, the paper will meet the quality and originality criteria to be published in Molecules.
Author Response
1. The only restriction of the scope of the reaction is the necessity to have an alkyl group on the nitrogen atom (typically a methyl group). The N-methyl deprotection seems to me not so easy especially with a metal free approach. As a minimum, some methods must be added as example from the literature. More interesting should be to synthesize a compound bearing a protecting group easier to remove such as a benzyl moiety. That will be a huge improvement in the quality of the paper.
Response: Benzyl protected aniline also underwent the reaction well to give the corresponding product in 82% yield (see product 24).
2. Some minor Errors and corrections must be addressed:
Line 31: one space is missing between “of” and “α”
Scheme 1. A. durgs => drugs
One carbon atoms is missing in the NMR description of the compounds 2 and 3.
There is a mistake in the 1H NMR integration value of two signals of the compound 2 at 3.84 and 2.44 ppm. That should be 3 instead of 4.
Response: All errors have been revised. The 13C NMR of compounds 2 and 3 has overlapping peaks.
Reviewer 2 Report
Xu, Tang and co-workers describe Bischler indole synthesis using HFIP under microwave condition. Indole is one of the most important structures in the pharmaceutical and agricultural chemistry. They employed HFIP as a promoter and showed good activity in the synthesis of indole. The yields of product are acceptable in Table 2. In addition, the proposed mechanism might be reasonable. Therefore, I recommend this manuscript to be published in Molecules. However, the followings will be addressed before the final acceptance.
The substrate without the N-methyl protecting group showed no product. How about the N-Boc or N-tosyl protecting group? Please add some examples of them.
How about the reaction with isopropanol instead of HFIP?
Author Response
1. The substrate without the N-methyl protecting group showed no product. How about the N-Boc or N-tosyl protecting group? Please add some examples of them.
Response: Benzyl protected aniline also underwent the reaction well to give the corresponding product in 82% yield (see product 24). The benzyl protecting group can be readily removed. We also studied the N-Boc protecting group, but it did not work.
2. How about the reaction with isopropanol instead of HFIP?
Response: The reaction with i-PrOH only gave product 2 in 21% yield.
Reviewer 3 Report
The synthetic methodology described is useful and hence this manuscript should eventually be published.
I am concerned about the lowermost part of Scheme 2, because all structures are wrong, thus “C2F6OH” shall be C3F6OH and the 2-position shall read CH2 not just CH. The best solution is to delete this part of the mechanism. The top-most mechanism is fine. In the general part (p.2) correct “CF3CH3OH” to CF3CH2OH.
When it comes to compound 2 the authors should mention that this molecule is produced in multi-ton quantities as a stabilizer of e.g. polymers. Delete the insignificant ref. [34].
Author Response
1. I am concerned about the lowermost part of Scheme 2, because all structures are wrong, thus “C2F6OH” shall be C3F6OH and the 2-position shall read CH2 not just CH. The best solution is to delete this part of the mechanism. The top-most mechanism is fine. In the general part (p.2) correct “CF3CH3OH” to CF3CH2OH.
Response: All errors have been corrected, see Scheme 2.
2. When it comes to compound 2 the authors should mention that this molecule is produced in multi-ton quantities as a stabilizer of e.g. polymers. Delete the insignificant ref. [34].
Response: Ref. [34] has been deleted.
Reviews for indole as inhibitors of tubulin polymerization were added. See:
[31] Brancale, A.; Silvestri, R. Indole, a core nucleus for potent inhibitors of tubulin polymerization. Medicinal Research Reviews 2007, 27, 209-238. DOI: 10.1002/med.20080.
[32] Sang, Y.-L.; Zhang, W.-M.; Lv, P.-C.; Zhu, H.-L., Indole-based, Antiproliferative Agents Targeting Tublin Polymerizaton. Current Topics In Medicinal Chemistry 2017, 17, 120-137. DOI: 10.2174/1568026616666160530154812